# Comparison of the Real Part of Dielectric Constants with Different Materials to Decrease the Emittance and a Virtual Dielectric Constant to Reproduce Reflectance

**Jesús Manuel Gutiérrez-Villarreal [1,2]**, **Horacio Antolin Pineda-León [3]**, **Mario F. Suzuki Valenzuela [1]**,
**Ian Sosa-Tinoco [1,*]** and **Santos Jesús Castillo [3]**

1   Electrical and Electronics Department, Sonora Institute of Technology, Obregon City 85000, Mexico;
    jmgutierrezv86@gmail.com (J.M.G.-V.); mario.suzuki@itson.edu.mx (M.F.S.V.)
2   Mechatronics Engineering Department, Technological University of Southern Sonora, Dr. Norman E. Borlaug
    Street, Obregon City 85190, Mexico
3   Physics Department, University of Sonora, Hermosillo 83000, Mexico; horanpile@hotmail.com (H.A.P.-L.);
    santos.castillo@unison.mx (S.J.C.)
*   Correspondence: ian.sosa@itson.edu.mx

**Abstract:** This paper analyzes how the real part of the dielectric constant affects the emittance or temperature in some materials. A two-layer configuration was implemented on a glass substrate, with theory and experiment, on a sunny day in Mexico. Furthermore, the transfer matrix method was used as theory, changing the material on the top of the substrate and below a film of zinc sulfide. As a result, for a larger real part of the dielectric constant, the emittance decreased in analytical results, and therefore a decrease in temperature was obtained in the experiment. Furthermore, a virtual dielectric constant was obtained from the experimental reflectance in a bilayer system reproducing this system analytically with one layer having different thickness. The finite-difference time-domain (FDTD) method was used to obtain the optimal length of equilateral pyramids on the surface of a flat film by changing the materials to improve the reflectance or decrease the emittance. It was concluded that for a wavelength of the incident source, optimal dimensions of the triangles on the surface exist.

**Keywords:** FDTD; passive cooling; reflectance

## 1. Introduction

One of the reasons to research passive cooling materials is to save electrical energy since a large amount of electric energy for refrigeration is used globally; for example, the global residential energy consumption is 26.6% [1]. In addition, due to their properties, passive cooling materials can help to reduce the heat island effect in cities on roofs and streets [2]. Radiative passive cooling, also known as radiative cooling or nighttime cooling, is a phenomenon and technology that utilizes the natural emission of thermal radiation to cool objects and surfaces without the need for external power sources or active cooling mechanisms. This process takes advantage of the fact that all objects with a temperature above absolute zero emit thermal radiation in the form of electromagnetic waves, primarily in the infrared (IR) spectrum.

The basic principle of radiative passive cooling involves allowing an object to radiate its heat energy into the cold outer space [3], where temperatures can be significantly lower. This is typically achieved by using materials and surfaces that are highly emissive in the infrared spectrum and have low absorptivity for solar radiation (sunlight) during the day. By emitting more thermal radiation than they absorb from the surroundings, these materials can achieve a net cooling effect.

The idea of this work arises from the development of passive radiative cooling materials, which consist of arrangements of materials to cool the surface temperature. In this train

of thought, Ref. [4] reported that the temperature of the last layer was 5 °C lower than that of the environment at the zenith. There have been various subsequent passive radiative cooling studies, such as the research by [5]. The authors reported a temperature drop of 6.2 °C against ambient temperature corresponding to a net cooling power of 19.7 W/m² in a non-vacuum configuration during the shaded peak day, and subsequently they designed a 8 μm triangular structure with silicon. The work reported with the greatest decrease in temperature was carried out by [6], in which the temperature of the underlying silicon absorber was reduced by as much as 13 °C. Finally, there is the work carried out by [7], which reported a reflectivity result of 98% in the solar spectrum in Hong Kong, which allows a temperature drop of up to 2.7 °C with a solar intensity of 1000 W/m² on a day with high humidity. In a related work, [8–12] designed, optimized, fabricated, and characterized highly reflective quasi-omnidirectional multilayer structures with a wide angular and spectral range.

It can be seen that research on passive radiative cooling focuses on the race to decrease the surface temperature and determine which materials radiate more energy. As well as passive radiative cooling, there are many methods related to minimizing the temperature of surfaces. There are some examples, such as the as floating tracking concentrating cooling system [13], hybrid solar photovoltaics [14], a thermal system cooled by water spraying [15], a hybrid solar photovoltaic/thermoelectric system [16], and other hybrid solar photovoltaics [17].

For that reason, this work focuses on answering the question: How does the real part of the dielectric constant affect the temperature on the surface of the proposed passive radiative material? It is evident that the imaginary part of this constant is related to the optical absorption of the material and thus contributes more to its surface temperature. What, then, is the impact of the real part? Furthermore, this research consists of obtaining a virtual dielectric constant to reproduce the experimental reflectance using a single equation. It is also evident that the greater the energy a material reflects, the lower the surface temperature, and the lower its emittance will be. Therefore, efforts are being made to obtain optimal measurements for pyramid shapes on the surface of material layers. Are there any optimal sizes of surface pyramids that depend on the incident wavelength? This research work focuses on answering these questions.

## 2. Emittance of Materials with Different Dielectric Constant

### 2.1. Theoretical Simulation of the Radiation Light

With the transfer matrix method (TMM) and TE polarization, normal light will be incident on the material arrangement surface as shown in Figure 1. We only worked with TE polarization due to the fact that the reflection or emittance (1-R) results are similar for both TE and TM polarization; see, for example, Figure 2. The TMM is used in optics and acoustics to analyze the propagation of electromagnetic or acoustic waves through a stratified medium [18]. The selected materials with different dielectric constants at room temperature are zinc sulfide (ZnS), silicon (Si), silicon dioxide ($SiO_2$), hafnium dioxide ($HfO_2$), selenide of plome (PbSe), and lead sulfide (PbS) [19].

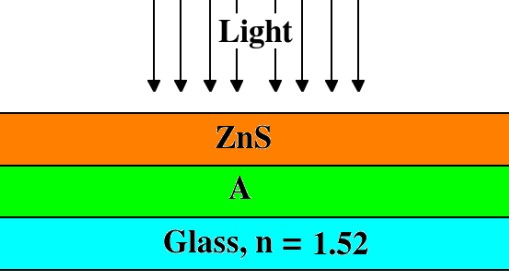

**Figure 1.** Scheme comparing the emittance of different bilayer arrangements. Material A is the material that can be substituted with Si, $SiO_2$, PbSe, PbS, and other materials.

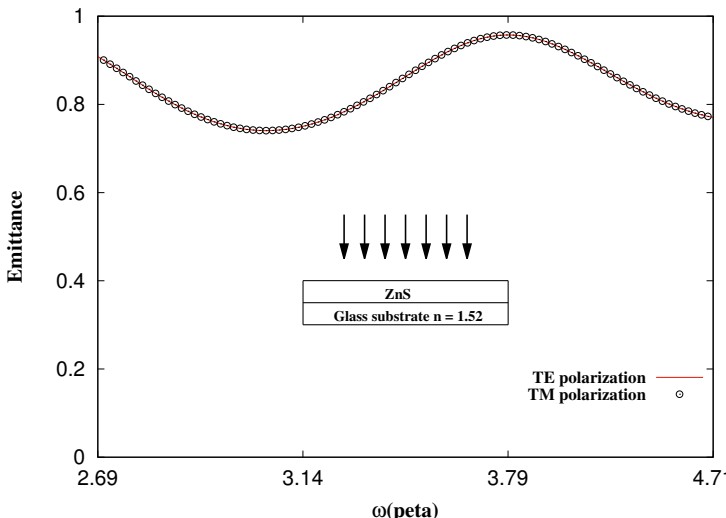

**Figure 2.** Emittance of light vs. frequency (rad/s) in the visible spectrum.

First, the real part of the dielectric constant of different materials is shown in Figure 3. The variation in permittivity with the wavelength in Figure 3 is primarily manifested in the real part of permittivity ($\varepsilon$). This is attributed to the fact that different wavelengths of electromagnetic radiation can interact with electrons and molecules within the material in distinct ways. As the wavelength of radiation changes, resonance and molecular interactions can also shift, consequently influencing the polarization of the material and its response to an applied electric field. When the real part of a material's permittivity exhibits a higher value, it signifies that the material tends to polarize more in response to an applied electric field in comparison to materials possessing a lower real part of permittivity. As previously mentioned, this permittivity variation occurs when the wavelength or frequency of the incident source change through the medium. The emittance of the materials with a different real part of the dielectric constant are compared in Figure 4. Furthermore, the emittance is presented in most cases and figures in this paper as it enhances the analytical approach to cooling analysis [20].

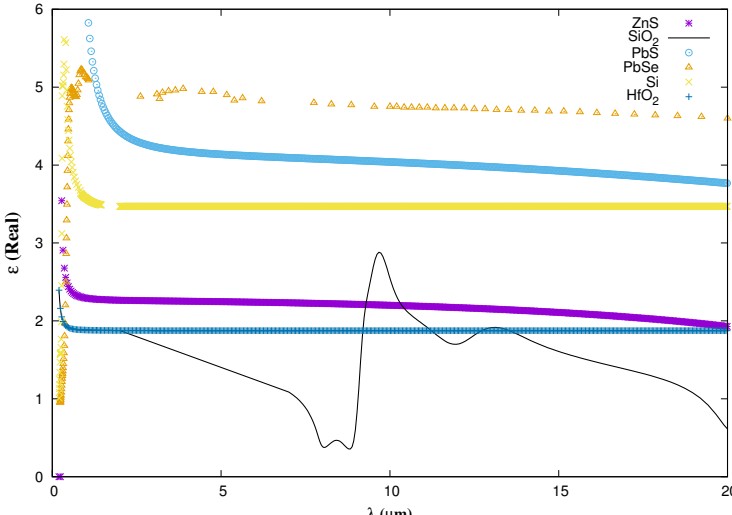

**Figure 3.** Real part of dielectric constants vs. wavelength at room temperature.

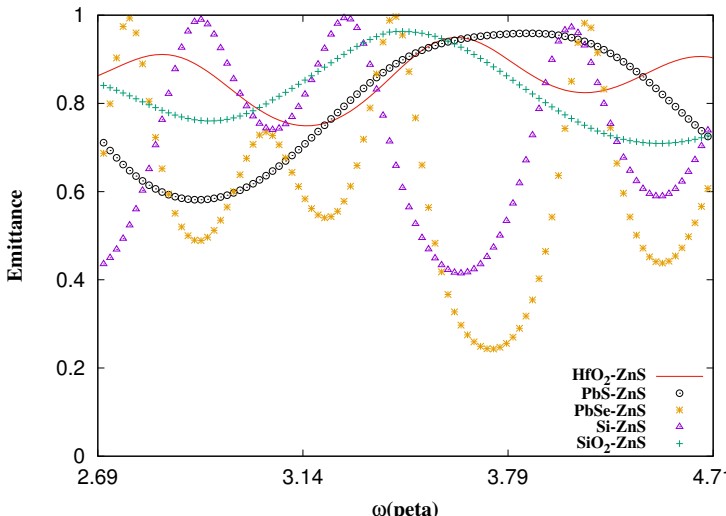

**Figure 4.** Emittance vs. frequency. Thickness of 300 nm of each layer. We used the configuration shown in Figure 1 to obtain the emittance.

The thickness of ZnS is 300 nm as with every material used detailed in this section. We applied the TMM to obtain the emittance of bilayer systems. Figure 2 shows the emittance of ZnS on the substrate in the visible spectrum. The substrate is a glass slide with a refractive index of 1.52.

From Figure 4, it is easy to observe that the larger the value of the real part of the dielectric constant, the lower the emittance. Figures 3 and 4 show that the material with the highest dielectric constant (PbSe) has the lowest emittance, without considering the absorption or the imaginary part of the said dielectric. Additionally, it is understood that the more absorption a material has, the lower its reflection or higher the emittance. By comparing the areas above the emittance curves for each material, Table 1 was generated to ensure that the largest area is indeed the PbSe curve.

**Table 1.** Total area of the emittance above the spectrophotometric response curve for each material.

| Material | Area above the Curve |
|---|---|
| PbSe-ZnS | 7.527 (a. u.)$^2$ |
| Si-ZnS | 5.576 (a. u.)$^2$ |
| ZnS | 3.457 (a. u.)$^2$ |
| HfO$_2$-ZnS | 2.989 (a. u.)$^2$ |
| SiO$_2$-ZnS | 2.42 (a. u.)$^2$ |
| PbS-ZnS | 1.29 (a. u.)$^2$ |

## 2.2. Sample Surface Temperature Experiment

The theoretical results were verified with a experimental setup with different samples of each material. The experiment was performed in Obregon City, Sonora, Mexico, on a sunny summer day. The temperature of the back of each material was measured with a type K thermocouple connected to a Keysight 34970A datalogger, and the irradiation was measured with a Kipp & Zonen Cmp3 pyranometer. A bilayer system was employed to mitigate the impact of color absorption from the outermost layer. This ensures uniformity in the surface color across all systems, particularly within the experimental context.

The samples of silicon (Si) wafers, silicon wafers with zinc sulfide (Si-ZnS), dioxide of silicon wafer (SiO$_2$), dioxide of silicon wafer with zinc sulfide (SiO$_2$-ZnS), dioxide of hafnium wafer (HfO$_2$), and dioxide of hafnium wafer with zinc sulfide (HfO$_2$-ZnS), in that order, were used to investigate the real part of the dielectric constant and compare it with the surface temperature of the material.

Figure 5 depicts the material samples used for the experiments, while Figure 6, presents the analysis of the temperature measurements with thermocouple devices are shown for each of the samples; it is to be noted that the samples with zinc sulfide recorded low temperatures. The following table shows the areas under the curve of each sample in Figure 5.

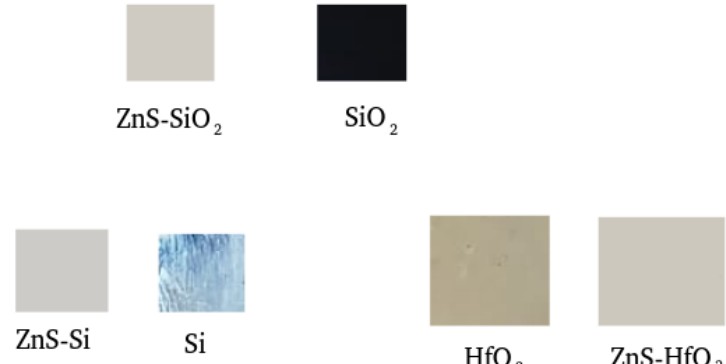

**Figure 5.** Samples of the materials used in surface temperature measurements with approximately equal thicknesses.

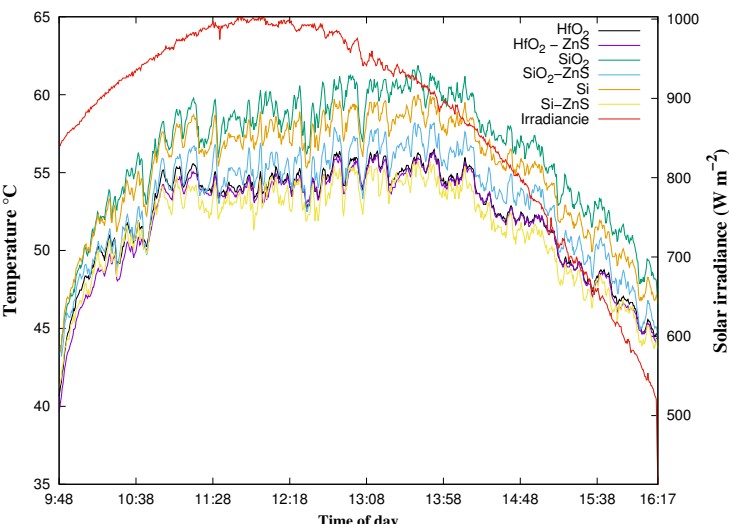

**Figure 6.** Temperature of the sunny day (15 July 2022) vs. the hours of the day. Left axis is graphed with the curve in red and is the irradiation of the day.

When comparing the table of area temperature versus the area emittance (Table 2), it is clear that the low emittance in the samples with zinc sulfide have low temperatures or a small area under the curve (theoretical). For example, in the area emittance table, the amount of Si-ZnS is 5.576 (a. u.)$^2$, being the second highest emittance, and this agrees with being the coldest sample in the area temperature table (experimental). Moreover, it coincides with the finding that, with lower emittance, the temperature decreases in the samples with zinc sulfide on the top. Once again, from the area temperature table, it is shown that pure silicon is hotter (greater area above the curve) than hafnium dioxide. The materials are affected by their color and absorption. Therefore, they should be in that order when adding the zinc sulfide layer above. However, they are not, and this is because the greater the real part of the dielectric constant of the material, the greater the energy that the material reflects outward, and of course the lower the temperature on the surface.

There are several methods for manufacturing thin films. In our case, we used screen printing to add a ZnS layer onto Si, $SiO_2$ and $HfO_2$ wafers as well as onto a glass substrate. To apply a layer of this material, it first had to be manufactured in the laboratory and then applied as a fresh coat of paint using the aforementioned silk-screen printing method. To ensure that the indicated materials were deposited onto the substrate and wafers, samples were taken and analyzed using X-ray photoelectron spectroscopy (XPS) equipment in order to confirm that the samples used in the experiment contained the materials discussed in this study, for example ZnS, hafnium, $SiO_2$, and Si (Figures 7–9).

**Table 2.** Total area of the temperature under the spectrophotometric response curve for each material.

| Material | Area under the Curve |
| --- | --- |
| Si-ZnS | 20,035.96 (a. u.)$^2$ |
| $HfO_2$-ZnS | 20,213.75 (a. u.)$^2$ |
| $SiO_2$-ZnS | 20,738.09 (a. u.)$^2$ |
| Si | 21,504.70 (a. u.)$^2$ |
| $HfO_2$ | 20,355.97 (a. u.)$^2$ |
| $SiO_2$ | 21,972.31 (a. u.)$^2$ |

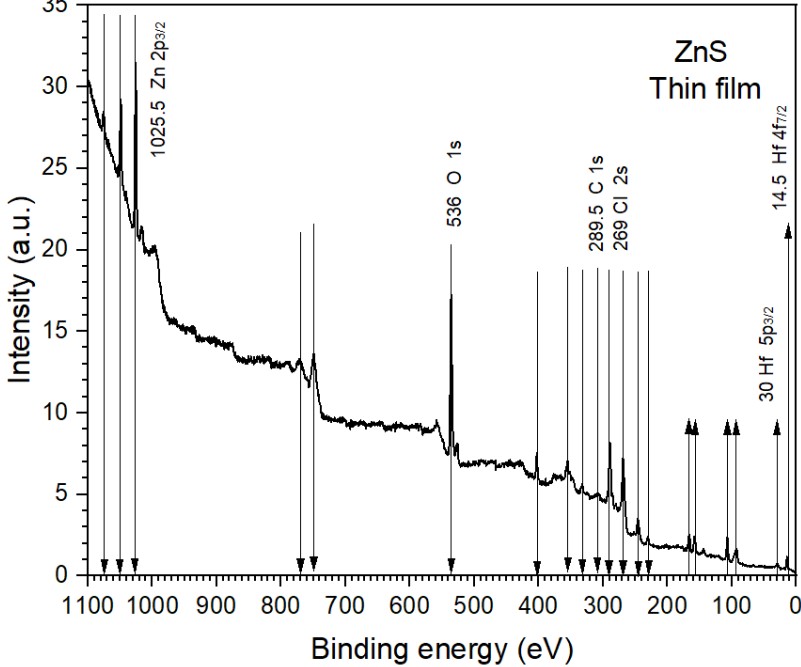

**Figure 7.** XPS characterization and X-ray photoelectron spectroscopy to show the materials in the substrate. In this case, ZnS is shown on the top and $HfO_2$ below.

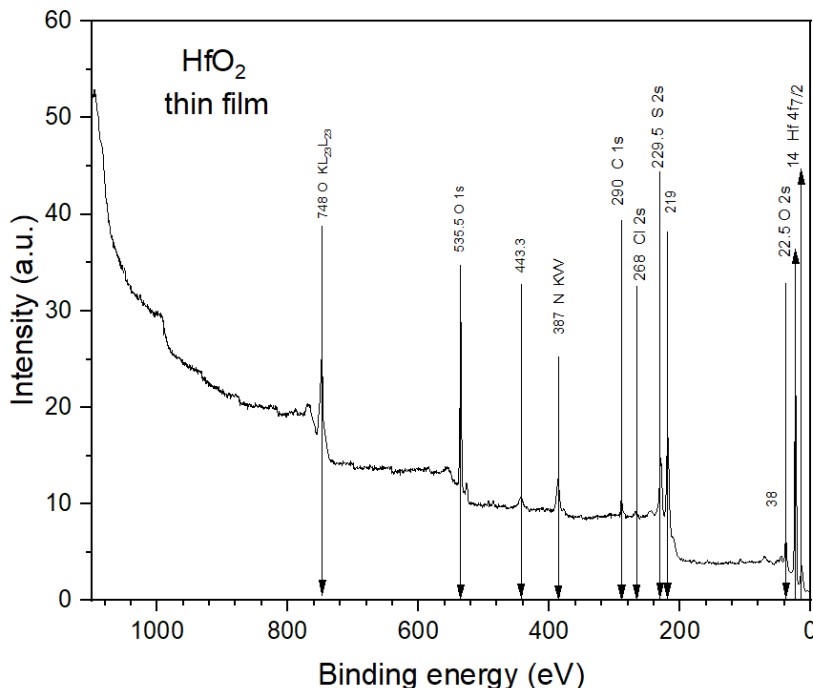

**Figure 8.** XPS characterization and X-ray photoelectron spectroscopy to show the materials in the substrate. In this case, $HfO_2$ thin film is shown.

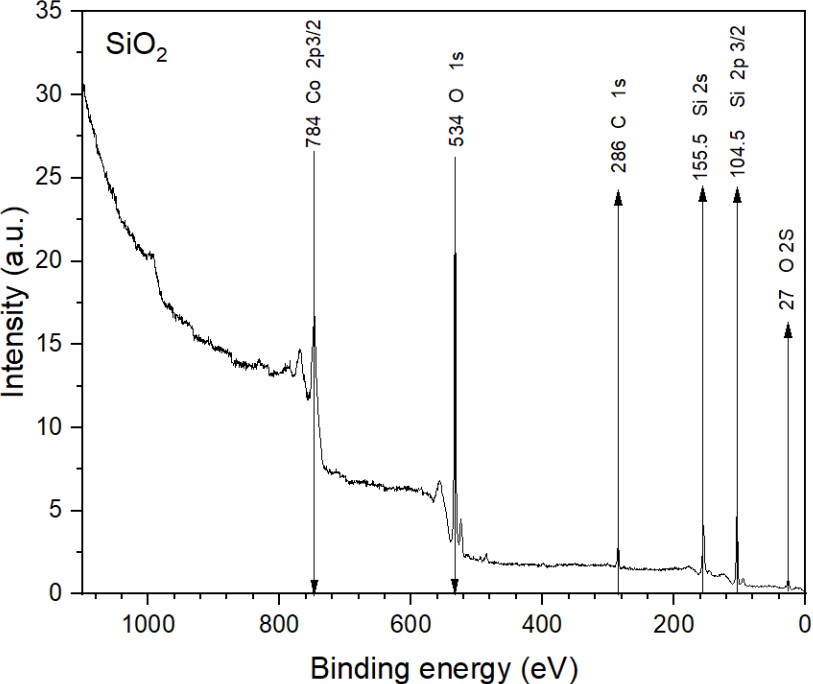

**Figure 9.** XPS characterization and X-ray photoelectron spectroscopy to show the materials in the substrate. In this case, $SiO_2$ thin film is shown.

### 2.3. Experiment Discussion

In this section, the experiments conducted in this work are described. A formulation for synthesizing zinc sulfide through the screen printing method is presented. This choice is driven by the fact that the chemical bath method resulted in an excessively thin surface layer. Therefore, an alternative approach was adopted to generate a thicker ZnS surface film.

The formulation for producing ZnS involves the following mixture:

1 mL of zinc acetate $Zn(O_2CCH_3)_2$;
1 mL of Thiourea $CH_4N_2S$.

These components are mixed at room temperature and then heated to 80 °C for 20 min. Figures 7–9 illustrate the materials within the samples.

Once the ZnS material is prepared, a paste is applied to a glass slide or substrate and left to dry for several hours, thus creating a ZnS film.

However, a challenge arose in measuring the temperature on the material's surface. Initially, a thermal camera was employed. However, obtaining accurate temperature measurements with this camera proved difficult due to the materials' varying emissivity with temperature and material radiation (which is not constant). Consequently, a design was pursued involving a round, flat metallic body that would make direct contact with the material's surface. This metallic body was connected to thermocouple cables to facilitate temperature sensing. Additionally, solar radiation was measured using a radiation sensor. The solar irradiance curve obtained bears a striking resemblance to curves observed in other studies [4]. It is important to acknowledge that the curve presented in this paper cannot be expected to perfectly match those from previous studies due to geographical variation.

## 3. Reflection of ZnS in the Visible Spectrum
### 3.1. Reflectance Experiment

The experiment consisted of obtaining the absorption of a ZnS film on a glass substrate like Figure 5 with ZnS 2 μm of thickness. The thickness of the substrate was 1 mm. We introduced the film to visible spectrum equipment and obtained the absorption ($A$). The equation for the transmission ($T$) of light through material media related to $A$ is

$$T = 10^{-A}, \tag{1}$$

with $A$, we obtain the transmittance $T$ from Equation (1). In addition, we achieve the reflectance with the next equations,

$$Log_{10}(T) = -A, \tag{2}$$

resulting in Equation (2) as:

$$Log_{10}(1/T) = A, \tag{3}$$

with $A$ and $T$, it is easy to obtain $R$ with the equation:

$$R = 1 - (A + T). \tag{4}$$

Figure 10 is the result of obtaining the absorption in the visible spectrum equipment.

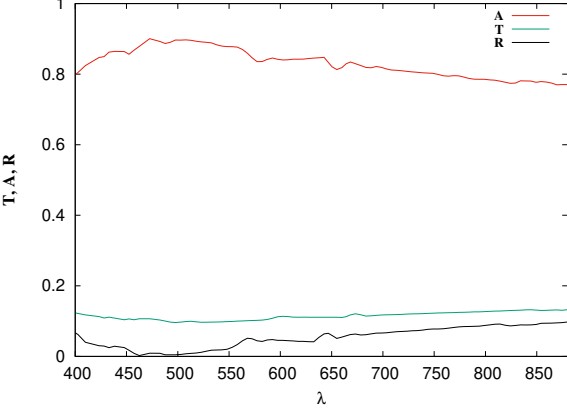

**Figure 10.** Reflection, absorption and transmission of ZnS on a substrate of glass vs. wavelength in nm with normal incidence.

### 3.2. Theoretical Reproduction of Reflectance

From the experiment, we obtained the data for $R$ for the reflectance depending on the wavelength, so we have Equation (5) for $R$ with normal incidence:

$$R = \frac{(n-1)^2 + k^2}{(n+1)^2 + k^2},$$

(5)

where $k$ is the extinction coefficient, and $d$ is the experimental thickness of the film, with 2 μm.

$$k = \frac{\lambda A}{4\pi d}.$$

(6)

It is necessary to obtain the refractive index $n$ from the previous equations. Solving for $n$, Equation (5) becomes

$$n = \frac{(R+1) \pm \sqrt{4R - k^2(R-1)^2}}{(1-R)}.$$

(7)

From Equation (7), only the positive part outside the radical is used since the negative part gives data for $n$ less than unity. With the previous data of $n$ and $k$, a virtual dielectric constant is attained to reproduce the experimental results, albeit now in an analytical way.

$$\epsilon = (n + ik)^2.$$

(8)

The theoretical reflectance of a three-layer system with TM and TE polarization is:

$$R = \frac{r_{0,1}r_{1,2}exp(2ik_yd)}{1 + r_{0,1}r_{1,2}exp(2ik_yd)}.$$

(9)

Equations (8) and (9) are employed for the analytical reproduction of $R$. (see Figure 11).

For Equation (9) $k_y$ is the wave vector with normal incidence in TM polarization. $r_{01}$ is the reflection from impinging media and the second respectively, and $d$ is the thickness of the film. The most important thing to note is that the experiment to obtain $R$ involved a ZnS layer on a glass substrate and the theory to obtain $R$ involved a virtual dielectric constant obtained from those layers considering only one new material; in addition, we obtained the same curve for $R$ with different thicknesses.

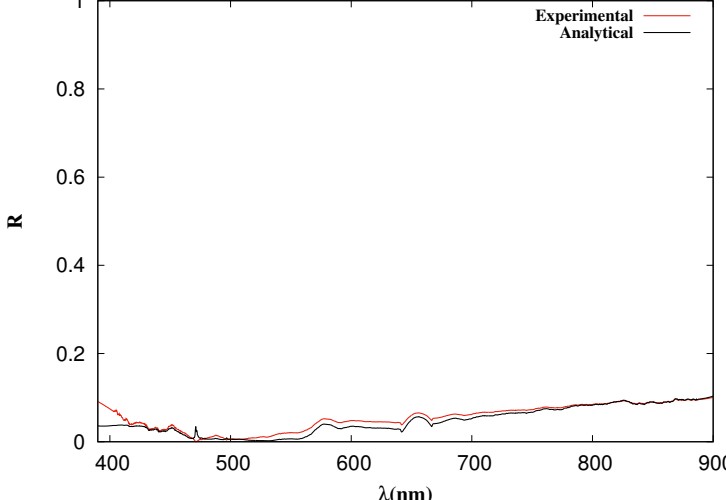

**Figure 11.** Reflection of a thin film vs. wavelength. The red line is the experimental R of ZnS film with 2 μm of thickness on a substrate. The black line is analytic reflection using a virtual dielectric constant with thickness of 110 nm; the impinging media was a glass of $n = 1.5$.

## 4. FDTD Emittance of a Flat Layer and Triangular Structures

The finite-difference time-domain (FDTD) method is a technique that discretizes the Maxwell equations for fixed geometries in contrast to TMM, which only works with planar surfaces. It can work with non-planar surfaces, as demonstrated in this study, where we used it to calculate the emittance from a planar surface and triangles placed above it, with different materials interchanged.

The work was carried out with a flat surface and seven equilateral pyramids to decrease the emittance. It is well known that triangular geometries offer better reflection and lower emittance. We determined the optimum sizes and lengths of these pyramids, as shown in Figure 12.

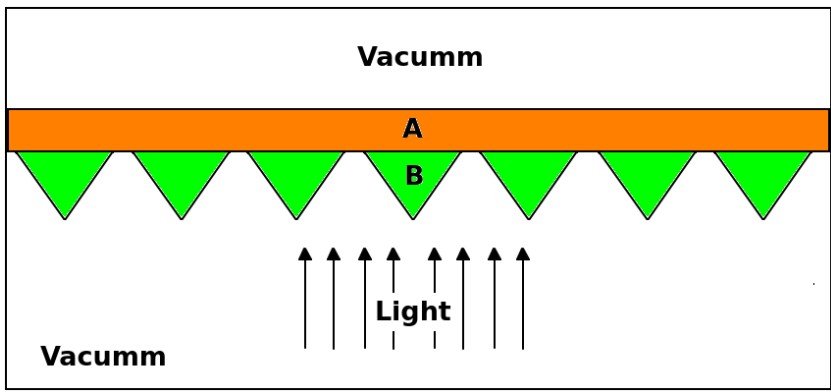

**Figure 12.** The flat A and pyramids B are interchangeable materials.

The partition in FDTD that we used is $\Delta x = 320$ nm/100; with this, we have an error tolerance of 1.8% compared with the analytical results [21].

The sides of the pyramids as well as the length of separation between them are equal. The field with TE polarization inside the FDTD (2D) impinging in the mentioned configuration is shown in the figure.

Figure 13 shows the analysis of the emittance of the impinging field, with the aim of determining an optimal length of the measures of the Si pyramids and ZnS flat with a thickness of 320 nm.

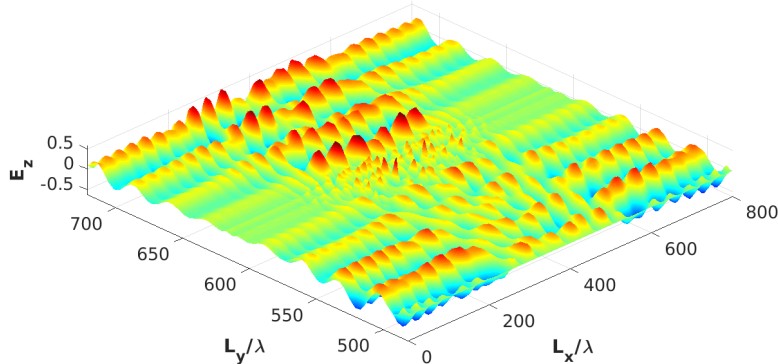

**Figure 13.** The field inside FDTD with a film of ZnS and pyramids of Si at the center. $L_y/\lambda = 600$, where $L_x$ and $L_y$ are lengths. The impinging beam is a pulse of 1.77 (Femto s).

It is observed from Figure 14 that there is a decrease in emittance with the measure of separation and sides of Si pyramids on a ZnS thin film with $L = 0.5\lambda$. The decrease in the emittance of Figure 14 and the next figures is the optimal length for the size of the triangles on the surface.

The emittance was then compared by altering the sequence of the ZnS film with that of Si, as well as the pyramids made of ZnS.

The combination of the materials between the flat film and the equilateral pyramids improves the reflection or decreases the emittance thanks to the triangular structure on the surface with optimal measures; however, there is one exception due to the absorption of "Si", which is very low compared with other materials. A flat film of "Si" emits less without adding triangular structures on the top surface.

From Figure 15, it is to be noted that the highest reflection occurs with the Si thin film alone, without the pyramids; this is due to the absorption of Si, which is very low compared to that of ZnS. The pyramidal structure does not always improve reflection or decrease emittance. Figure 16 shows the reflection of a thin film (thickness = 320 nm) of HfO₂ with ZnS pyramids, it is observed that the size of pyramids where reflection is improved is still $1/2\lambda$.

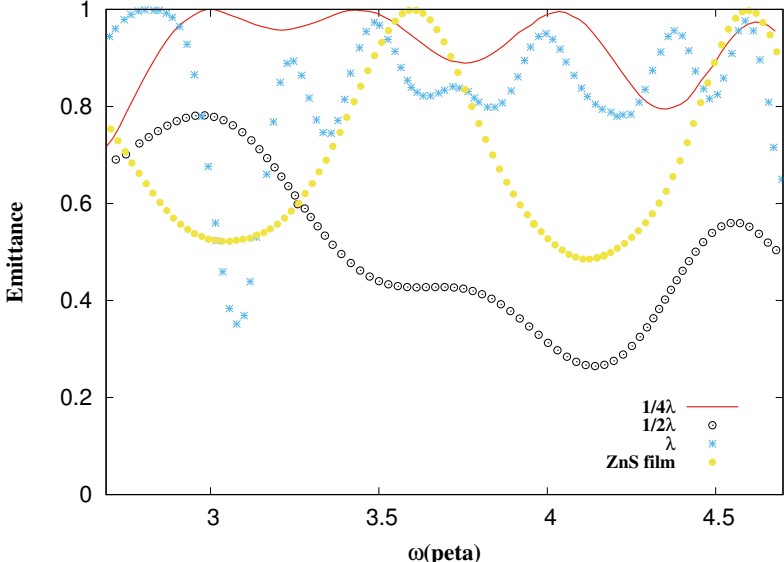

**Figure 14.** Emittance vs. frequency. The black circles are the emittance from a thin film of ZnS and pyramids of Si with $L = 0.5\lambda$, the red line is from pyramids of size $L = 0.25\lambda$, the blue points are from $L = 1\lambda$, and the yellow points are from the ZnS thin film alone.

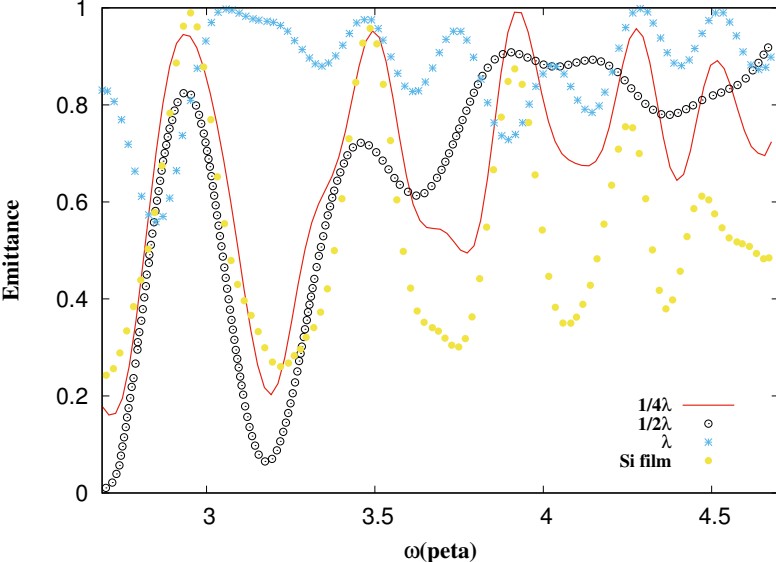

**Figure 15.** Emittance vs. frequency. The black circles are the emittance from a thin film of Si and pyramids of ZnS with $L = 0.5\lambda$, the red line is from pyramids of size $L = 0.25\lambda$, the blue points are from $L = 1\lambda$, and the yellow points are from the Si thin film alone.

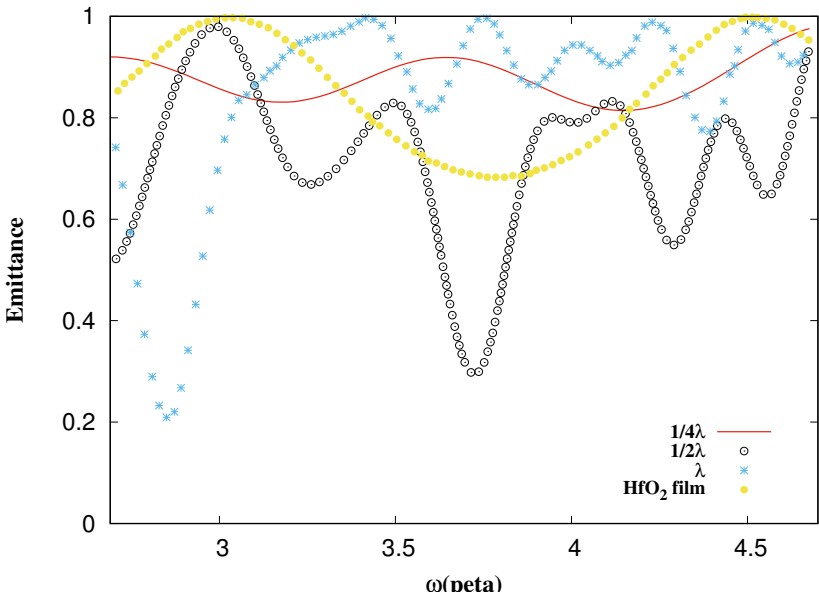

**Figure 16.** Emittance vs. frequency. The black circles are the emittance from a thin film of HfO$_2$ and pyramids of ZnS with $L = 0.5\lambda$, the red line is from pyramids of size $L = 0.25\lambda$, the blue points are from $L = 1\lambda$, and the yellow points are from the HfO$_2$ thin film alone.

Once again, it can be confirmed that the emittance decreases with the pyramid measure of $1/2\lambda$ (see Figure 17). It is observed that it is very similar to Figures 18 and 19.

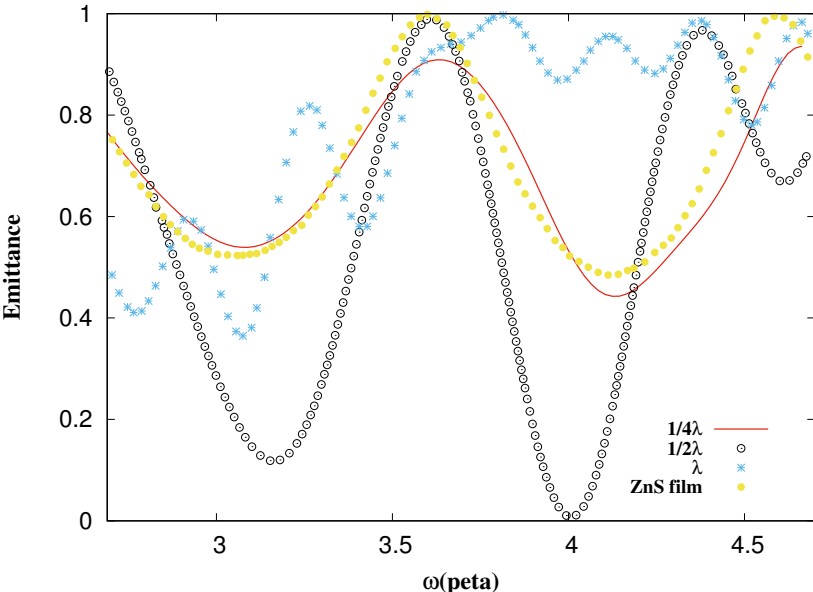

**Figure 17.** Emittance vs. frequency. The black circles are the emittance from a thin film of ZnS and pyramids of HfO$_2$ with $L = 0.5\lambda$, the red line is from pyramids of size $L = 0.25\lambda$, the blue points are from $L = 1\lambda$, and the yellow points are from the ZnS thin film alone.

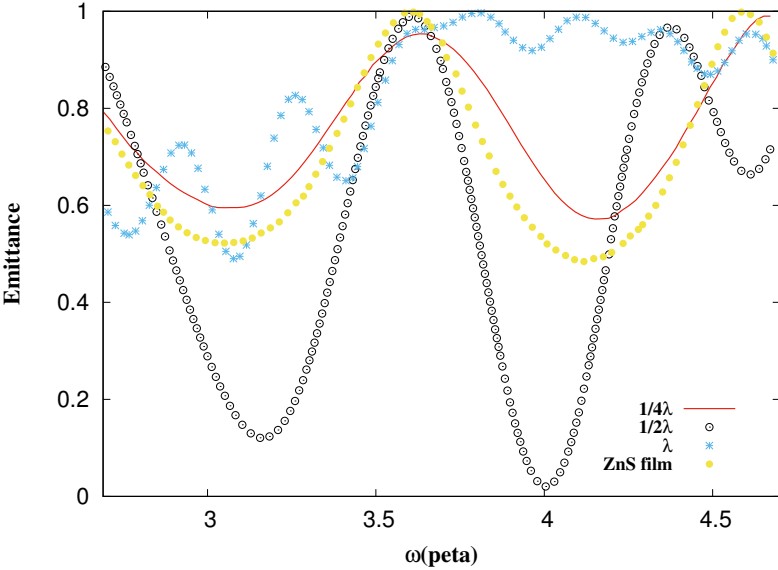

**Figure 18.** Emittance vs. frequency. The black circles are the emittance from a thin film of ZnS and pyramids of SiO$_2$ with $L = 0.5\lambda$, the red line is from pyramids of size $L = 0.25\lambda$, the blue points are from $L = 1\lambda$, and the yellow points are from the ZnS thin film alone.

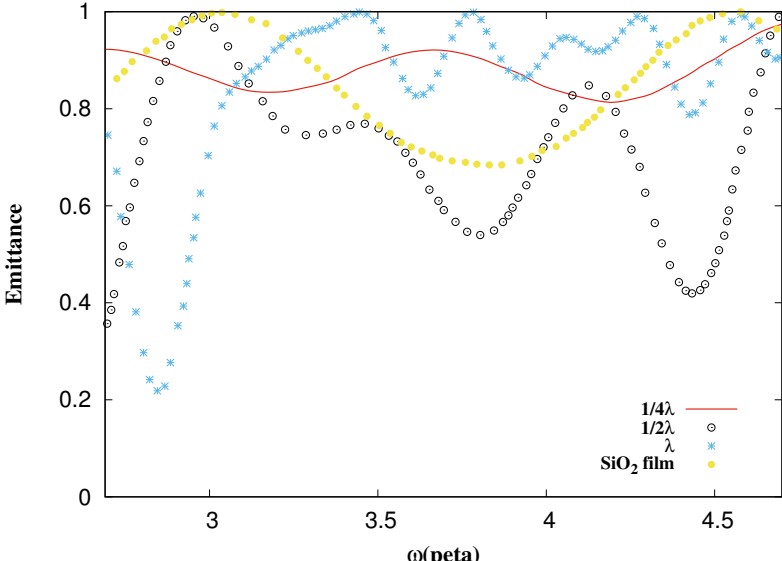

**Figure 19.** Emittance vs. frequency. The black circles are the emittance from a thin film of SiO$_2$ and pyramids of ZnS with $L = 0.5\lambda$, the red line is from pyramids of size $L = 0.25\lambda$, the blue points are from $L = 1\lambda$, and the yellow points are from the SiO$_2$ thin film alone.

## 5. Conclusions

The investigated materials in this research displayed a notable influence on light reflection and energy emissivity due to variation in their real dielectric constants. The outcomes were validated through a combination of theoretical models and empirical investigations, consistently indicating that a heightened part real of the dielectric constant corresponds to diminished material emissivity. Consequently, an increased real part of the dielectric constant contributes significantly to the moderation of material surface temperature by means of reflection or decreased emissivity. However, it is important to note that this effect does not necessarily surpass the impact of the imaginary component. Nonetheless, it undeniably plays a role in lowering surface temperature. These findings bear promising implications for applications demanding effective heat dissipation, such as thermal

insulation technologies and passive radiative cooling systems. An additional notable application of this study resides in the realm of solar cell design, where maintaining a cooler surface temperature is paramount to sustaining optimal electrical generation efficiency. Moreover, a future inquiry into the bilayer material arrangement entails the measurement of potential differences generated within the layers. This assessment seeks to establish whether alterations in the part real of dielectric constant lead to an increase or decrease in electric potential. This valuable insight holds significance for the design of various electronic components including transistors and diodes.

Furthermore, a virtual dielectric permittivity was formulated to replicate experimental reflectance outcomes at a consistent thickness. This virtual permittivity proves instrumental in deciphering the optical properties of the material while circumventing the necessity for experimental reflection techniques. This innovation marks a pioneering advancement in the realm of physical sciences or a succinct yet impactful contribution to the discipline.

In addition, an exploration of optimal pyramid dimensions was undertaken to induce diminished emissivity within the surfaces of the materials with the visible spectrum. The meticulously selected separation dimension, along with equilateral pyramid sides measuring 160 nm or $1/2\lambda$ (where $\lambda = 320$ nm), were demonstrated as optimal. When a bilayer material configuration boasts a minimally absorptive surface, it inherently reflects a greater amount of energy, thereby resulting in a pronounced reduction in surface temperature. Consequently, configurations with a low-absorption material placement yield heightened light reflection, irrespective of the pyramid surface structure. The pyramid-like surface architecture amplifies reflection within the material layers, particularly when tailored optimally for the intended source wavelength. Hence, for each wavelength of radiation, there exist optimal sizes of surface pyramids.

**Author Contributions:** Conceptualization, J.M.G.-V. and H.A.P.-L.; methodology, J.M.G.-V., H.A.P.-L., M.F.S.V., I.S.-T. and S.J.C.; software, J.M.G.-V.; validation, J.M.G.-V., M.F.S.V. and I.S.-T.; formal analysis, J.M.G.-V., I.S.-T. and S.J.C.; investigation, J.M.G.-V., H.A.P.-L. and S.J.C.; resources, J.M.G.-V., I.S.-T. and S.J.C.; data curation, J.M.G.-V.; writing—original draft preparation J.M.G.-V., H.A.P.-L., M.F.S.V. and S.J.C.; writing—review and editing, I.S.-T. All authors have read and agreed to the published version of the manuscript.

**Funding:** The authors would like to thank CONAHCYT Mexico for financial support. We would also like to thank ITSON's Programa de Fomento y Apoyo a Proyectos de Investigación (PROFAPI-2023-127) for funding provided for this study.

**Institutional Review Board Statement:** Not applicable

**Informed Consent Statement:** Not applicable

**Data Availability Statement:** Data underlying the results presented in this paper are not publicly available at this time but may be obtained from the authors upon reasonable request.

**Conflicts of Interest:** The authors declare no conflicts of interest.

## Abbreviations

The following abbreviations are used in this manuscript:

| | |
|---|---|
| TMM | Transfer matrix method |
| FDTD | Finite-difference time-domain |

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
