# Peer review of "Comparison of the Real Part of Dielectric Constants with Different Materials to Decrease the Emittance and a Virtual Dielectric Constant to Reproduce Reflectance"

_photonics, doi:10.3390/photonics10090994_

Round 1

Reviewer 1 Report

In this paper the authors show the considerations on the materials that can have an impact on the reflection of light or the energy emittance. They made both theoretical/simulation based studies as well as experiments. This makes the paper quite consistent, especially in terms of conlusions formed. The data presented are interesting and worth to be published.

However, authors need to analyze some suggestions/remarks from my side to make the paper more comprehensive and easier to read/understanding.

1) Introduction is very general and the background for the authors' work is not presented in a clear form. Please try to expand the introduction.

2) Please try to omit personal pronouns (we, they, etc.) throughout the text. Scientific texts do not use such the forms.

3)  When authors describe the materials used in experiment (Section 2.2.) it is suggested to give chemical structural formulas for all materials - to be consistent with figures.

4) I do not understand the reason of placing Fig. 6 in the place where it is now. If these are the appearance of real samples used in experiment they should be presented at the beggining of the Section.

5) Conclusions should summarize the results not describe what was done in the paper. Please carefully consider rewrite the conclusions to show main findings from the work. In addition, please give some suggestions (also in Conclusions Section) how the results obtained may be applied in engineering practice.

6) It is suggested also to place the figure such as they do not cut the text. Incorrect placing concerns for example figs. 11, 12, 15, 16 i many many more. 

7) Please prepare references in the style accepted by Energies journal.

English looks OK. It is suggested only to omit personal pronouns (we, they, etc.) throughout the text.

Author Response

Dear reviewer, 

We tried to answer all your questions and concerns in the pdf file. Thanks a lot for taking the time to review our manuscript. 

Reviewer 2 Report

In this paper, the affects of real part of the dielectric constant on emittance or temperature are discussed. The authors theoretically provide a relatively reasonable explanation. The data are presented logically. This paper can be accepted after some minor questions.

1. Details on writing need to be fixed. For example, Line 3.

2. Please address any thickness variations of the layers.

3. Please address the reason why the dielectric constant in Fig 3 have wavery curve.

Author Response

Dear Reviewer #2,

I wanted to express my sincere gratitude for taking the time to review my manuscript. Your thoughtful insights and constructive feedback have been invaluable in refining the content and strengthening the overall quality of the paper.
